# Brain Iron Deficiency Changes the Stoichiometry of Adenosine Receptor Subtypes in Cortico-Striatal Terminals: Implications for Restless Legs Syndrome

**DOI:** 10.3390/molecules27051489

**Published:** 2022-02-23

**Authors:** Matilde S. Rodrigues, Samira G. Ferreira, César Quiroz, Christopher J. Earley, Diego García-Borreguero, Rodrigo A. Cunha, Francisco Ciruela, Attila Köfalvi, Sergi Ferré

**Affiliations:** 1CNC-Center for Neuroscience and Cell Biology of Coimbra, University of Coimbra, 3004-504 Coimbra, Portugal; matildesgcr@gmail.com (M.S.R.); carsamira@gmail.com (S.G.F.); rcunha@fmed.uc.pt (R.A.C.); akofalvi@gmail.com (A.K.); 2Integrative Neurobiology Section, National Institute on Drug Abuse, Baltimore, MD 21224, USA; cesar.quiroz-molina@nih.gov; 3Department of Neurology, Johns Hopkins University, Baltimore, MD 21224, USA; cearley@jhmi.edu; 4Sleep Research Institute, 28036 Madrid, Spain; dgb@iis.es; 5Faculty of Medicine, University of Coimbra, 3004-504 Coimbra, Portugal; 6Pharmacology Unit, Department of Pathology and Experimental Therapeutics, Faculty of Medicine and Health Sciences, Institute of Neurosciences, University of Barcelona, 08907 L’Hospitalet de Llobregat, Spain; fciruela@ub.edu; 7Neuropharmacology and Pain Group, Neuroscience Program, Institut d’Investigació Biomèdica de Belvitge, Idibell, 08907 L’Hospitalet de Llobregat, Spain

**Keywords:** adenosine A_1_ receptor, adenosine A_2A_ receptor, restless legs syndrome, brain iron deficiency, striatum, cortico-striatal terminals, thalamo-striatal terminals

## Abstract

Brain iron deficiency (BID) constitutes a primary pathophysiological mechanism in restless legs syndrome (RLS). BID in rodents has been widely used as an animal model of RLS, since it recapitulates key neurochemical changes reported in RLS patients and shows an RLS-like behavioral phenotype. Previous studies with the BID-rodent model of RLS demonstrated increased sensitivity of cortical pyramidal cells to release glutamate from their striatal nerve terminals driving striatal circuits, a correlative finding of the cortical motor hyperexcitability of RLS patients. It was also found that BID in rodents leads to changes in the adenosinergic system, a downregulation of the inhibitory adenosine A_1_ receptors (A_1_Rs) and upregulation of the excitatory adenosine A_2A_ receptors (A_2A_Rs). It was then hypothesized, but not proven, that the BID-induced increased sensitivity of cortico-striatal glutamatergic terminals could be induced by a change in A_1_R/A_2A_R stoichiometry in favor of A_2A_Rs. Here, we used a newly developed FACS-based synaptometric analysis to compare the relative abundance on A_1_Rs and A_2A_Rs in cortico-striatal and thalamo-striatal glutamatergic terminals (labeled with vesicular glutamate transporters VGLUT1 and VGLUT2, respectively) of control and BID rats. It could be demonstrated that BID (determined by measuring transferrin receptor density in the brain) is associated with a selective decrease in the A_1_R/A_2A_R ratio in VGLUT1 positive-striatal terminals.

## 1. Introduction

Restless legs syndrome (RLS) is a common sensorimotor disorder, whose basic components include a primary sensory experience, akathisia (an urgent need to move), and in about 80% of patients a secondary motor component, periodic leg movements during sleep (PLMS), can be found [1,2]. It has been postulated that the overlying framework of the disease is a biological bias toward maintaining alertness even in the face of severe sleepiness [3]. It is generally accepted that brain iron deficiency (BID) is one of the primary pathophysiological mechanisms in RLS [2,3,4]. In fact, BID in rodents recapitulates key neurochemical changes reported in RLS patients and shows an RLS-like behavioral phenotype [3,4]. These neurochemical changes include presynaptic hyperdopaminergic and hyperglutamatergic states [2,3,4]. Utilization of the BID-rodent model of RLS could allow the identification of new neurochemical pathways and changes not previously reported in the human condition. Alterations in the adenosinergic system found in the BID rodent model [3,5] was an important new translational finding that led to a new potential treatment for RLS [6,7].

More specifically, BID in rodents was found to differentially modify the levels of the main adenosine receptor subtypes in the brain, downregulating and upregulating the levels of A_1_ receptors (A_1_Rs) and A_2A_ receptors (A_2A_Rs), respectively [8,9]. The changes in A_1_R density were found both in the cortex and striatum and occurred under a less severe iron-deficient (ID) diet, as compared with the changes in A_2A_R density, which only occurred with a more severe ID diet [9]. The generalized decrease in A_1_R density was then interpreted as a hypoadenosinergic state, and it was hypothesized to account for both the reported hyperdopaminergic and hyperglutamatergic states in RLS [3,5]. In fact, it is well established that A_1_Rs mediate the universal adenosine-mediated inhibitory control of glutamatergic transmission in the brain [10]; A_1_Rs are also indirectly and directly involved in the adenosine-mediated brake of dopaminergic transmission [11,12]. A generalized A_1_R-dependent hypoadenosinergic state could provide the basis for the sensory-motor symptoms of RLS (akathisia and PLMS) and, since A_1_Rs are involved in the homeostatic sleep mechanism [13], could also account for the biological bias toward alertness/arousal seen in RLS.

A specific correlative finding of the hyperglutamatergic state of RLS was the demonstration in the BID-rodent model of an increased sensitivity of cortical pyramidal cells to release glutamate by their efferent striatal nerve terminals [14]. The functional isolation of these terminals, using an in vivo optogenetic-microdialysis technique, demonstrated that they are targets of drugs with therapeutic efficacy in RLS, including the dopaminergic compounds pramipexole and ropinirole and the α_2_δ ligand gabapentin [2,15]. The three compounds significantly counteracted glutamate release by cortico-striatal terminals from rats with BID [14], and their efficacy most probably depends on the presence of presynaptic inhibitory dopamine receptors (D_2_ and D_4_ subtypes) and voltage-dependent calcium channels expressing α2δ units [14,16].

Importantly for the adenosine perspective, striatal glutamatergic terminals are also endowed with A_1_Rs and A_2A_Rs, which we previously demonstrated to form A_1_R-A_2A_R complexes (heteromers) [17]. It was proposed that these A_1_R-A_2A_R heteromers provide a mechanism to fine-tune modulation of striatal glutamate release, whereby low concentrations of adenosine would preferentially activate A_1_Rs, promoting inhibition of glutamate release, while high concentrations would activate A_2A_Rs. As occurs in different brain glutamatergic synapses, the activation of A_2A_Rs would promote glutamate release by allosterically counteracting A_1_R activation in the A_1_R-A_2A_R heteromer and directly through adenylyl cyclase signaling [17,18,19]. A recently reported functionally important property of the A_1_R-A_2A_R heteromer is the loss of constitutive activity of the A_2A_R [20]. We therefore hypothesized that changes in the density of A_1_R and A_2A_R in cortico-striatal glutamatergic terminals could be involved in the BID-induced increased activity of cortico-striatal terminals. Downregulation of A_1_Rs should lead to a decrease in the ability of low basal concentrations of adenosine to tonically inhibit glutamate release, but also to an increase in the population of A_2A_Rs not forming heteromers and, therefore, showing constitutive activity, a mechanism that would be more relevant upon concomitant upregulation of A_2A_Rs.

That changes in A_1_R/A_2A_R stoichiometry in favor of A_2A_Rs could be involved in the BID-induced increased sensitivity activity of striatal terminals was supported by a recent study that also used the in vivo optogenetic-microdialysis technique. It was first shown that an A_1_R antagonist, which counteracts the activation of A_1_Rs by endogenous basal levels of adenosine, increases the sensitivity of cortico-striatal terminals to release glutamate [21]. In addition, increasing the extracellular levels of adenosine by application of dipyridamole, an inhibitor of equilibrative nucleoside transporters ENT1 and ENT2, inhibited basal and optogenetically-induced glutamate release by cortico-striatal terminals from rats with BID and controls [21]. These preclinical results predicted a possible clinical role of dipyridamole in RLS, which we could confirm in two recent clinical studies [6,7]. However, the actual BID-induced change in the A_1_R/A_2A_R stoichiometry in cortico-striatal terminals still needs to be demonstrated. In the present study we directly address this question by using a newly developed FACS-based synaptometric analysis to compare the relative abundance on A_1_Rs and A_2A_Rs in cortico-striatal glutamatergic terminals of control and BID rats.

## 2. Results

### 2.1. Diet-Induced Anemic Phenotype and Bid

As shown in Table 1, rats fed with the iron deficient diet (n = 11) showed a classical anemic phenotype, including a 16% reduction in body weight, severe reduction in red blood cell- and hemoglobin-related parameters, hypolymphocytemia, thrombocytopenia and a 91% reduction in serum iron content, as compared with controls (n = 10). The concomitant decrease in the level of iron in the brain can be determined by directly measuring the iron content or indirectly, as here documented, by analyzing the density or expression of the transferrin receptor, (TrfR), which is specifically upregulated with chronic cellular iron deficiency [22,23,24].

Thus, it has been experimentally demonstrated that in most areas of the rat brain there is a strong inverse correlation between iron content and TrfR protein density or mRNA expression, indicating that these can be used as indirect measures of BID in rodents [23]. BID was confirmed by a significantly increased density of transferrin receptors in their cortex. As Figure 1 documents, we found that TrfR density was statistically significantly increased in the cerebral cortex of the rats fed with the iron deficient diet by 111.9 ± 40.3% (n = 7–8; *p* < 0.05; two-tailed unpaired Student’s *t*-test with Welch’s correction).

### 2.2. Flow-Synaptometric Analysis of Striatal Nerve Terminals

Working with isolated striatal nerve terminals allows for the evaluating of the presynaptic density of adenosine receptors without the masking effect of high post-synaptic adenosine receptor levels, which is often a problem in microscopy (Figure 2A,B; see Section 4). Furthermore, this approach permits a robust and sensitive quantitative analysis of the colocalization of presynaptic markers [25]. First, we evaluated the frequency of striatal terminals (positive for synaptophysin; SYN^+^) that were positive for vesicular transporter VGLUT1 (VGLUT1^+^), VGLUT2 (VGLUT2^+^), A_1_R (A_1_R^+^) or A_2A_R (A_2A_R^+^) and the total A_1_R^+^/A_2A_R^+^ ratio. VGLUT1^+^ and VGLUT2^+^ striatal terminals correspond to cortico-striatal and thalamo-striatal glutamatergic terminals, respectively [26,27,28]. The frequency of presynaptic A_2A_R labeling was about half of what we previously observed in striatal nerve terminals of CD-1 mice and Wistar rats [25], therefore most probably representing an underestimation. Unfortunately, from several other commercial antibodies, only the Nittobo/Frontiers anti-A_2A_R antibody showed sufficient selectivity in our assay when comparing its labeling to striatal nerve terminals of wild-type and A_2A_R KO mice (see Section 4). The most likely explanation for this discrepancy is that the Nittobo/Frontiers anti-A_2A_R antibody was raised against mouse A_2A_Rs and it is not equally sensitive for different rat strains (M. Watanabe, personal communication). Nevertheless, a decreased sensitivity in the detection of A_2A_Rs should not affect the conclusions related to a BID-dependent alteration of the frequency of receptors in the striatal glutamatergic terminals.

As Figure 2C demonstrates, BID did not affect the relative abundance of glutamatergic terminals in the striatum. Notably, the ratio of VGLUT1^+^ and VGLUT2^+^ terminals was comparable to what one can infer from electron microscopy studies [28]. There was a tendency for a decreased frequency of A_1_R labelling (Figure 2D_1_) and an increased A_2A_R labelling (Figure 2D_2_) for the total synaptosomal population, but without reaching statistical significance. Nevertheless, when the ratio of A_1_Rs and A_2A_Rs labeling frequencies were calculated within the same animals, there was a marked and statistically significant reduction in the A_1_R^+^/A_2A_R^+^ ratio in the BID group (Figure 2D_3_).

Subsequently, we calculated the labelling frequency of A_1_Rs and A_2A_Rs in cortico-striatal (VGLUT1^+^) nerve terminals. Figure 3 (panels A_1_–A_3_) show representative dot plots for dual-labelled synaptosomes. In synaptosomes from rats with BID, the apparent reduction in the percentage of A_1_R colocalization with VGLUT1 and the increase in the percentage of A_2A_R colocalization with VGLUT1 were not statistically significant (Figure 3, panels B_1_ and B_2_), while the A_1_R^+^/A_2A_R^+^ ratio in VGLUT1^+^-terminals was significantly decreased (Figure 3, panel B_3_). On the other hand, BID did not significantly alter either the percentage of A_1_R^+^ or A_2A_R^+^ in thalamo-striatal (VGLUT2^+^) nerve terminals or the A_1_R^+^/A_2A_R^+^ ratio in those terminals (Figure 4).

## 3. Discussion

The present results provide a mechanistic explanation for the previously reported increased sensitivity of cortico-striatal glutamatergic terminals in rodents with BID [14,21]: a change in A_1_R/A_2A_R stoichiometry in favor of A_2A_Rs. Taking into account the technical limitations imposed by the relative low sensitivity of the A_2A_R antibodies, this change in A_1_R/A_2A_R stoichiometry indicates that it should lead to a decrease in the population of A_2A_Rs forming heteromers with A_1_Rs in those glutamatergic terminals [17]. A_2A_Rs have a significant constitutive activity which is blunted upon heteromerization with A_1_Rs [20]. Our theory is that A_2A_Rs, freed from the inhibitory control of A_1_Rs, recover their constitutive activity, which plays a major role in the BID-induced increased excitability of the cortico-striatal glutamatergic terminals.

We have recently found evidence for a predominant heterotetrameric structure of three different striatal A_2A_R heteromers, constituted by A_2A_R homodimers and either A_1_R, dopamine D_2_ receptor (D_2_R) or cannabinoid CB_1_ receptor (CB_1_R) homodimers [19,20,29]. A_2A_R-D_2_R heteromers are localized postsynaptically, in striato-pallidal GABAergic neurons [11,30], while A_2A_R-CB_1_R heteromers are also localized in the cortico-striatal terminal [20,25]. It has been recently shown that, different from the A_2A_R-A_1_R heteromers, the A_2A_R in the A_2A_R-CB_1_R heteromer preserves its constitutive activity [20]. Results of the same study suggest that the well-established cannabinoid-induced inhibition of striatal glutamate release can mostly be explained by a CB_1_R-mediated counteraction of the A_2A_R-mediated constitutive activation of adenylyl cyclase in the A_2A_R-CB_1_R heteromer [20]. A relative increase in the population of A_2A_Rs not forming heteromers with A_1_Rs would also favor a relative increase of A_2A_Rs forming heteromers with CB_1_Rs, which would also be expected to increase the population of A_2A_Rs with constitutive activity.

Although it was initially thought that A_1_R-A_2A_R heteromers permit a fine-tune modulation by adenosine, by which low and high concentrations of adenosine preferentially activate the signaling of A_1_Rs and A_2A_Rs, respectively (see Introduction and [17]), recent results from optogenetic-microdialysis experiments with the ENT1/ENT2 inhibitor dipyridamole question this hypothesis. Thus, the direct striatal administration of dipyridamole significantly decreased basal levels of glutamate and counteracted the optogenetic-induced glutamate release by cortico-striatal terminals in both rats with BID and naïve controls [21]. According to the initial hypothesis of operation of the A_1_R-A_2A_R heteromer, the opposite effect, an increase in glutamate release, should have been expected if adenosine had reached the optimal extracellular concentration to activate the A_2A_Rs. In fact, results also obtained with in vivo microdialysis experiments showed that exogenously added selective A_2A_R agonists, such as CGS21680, induce glutamate release in the striatum [31,32] by directly activating presynaptic A_2A_Rs in glutamatergic terminals [33,34]. Given dipyridamole’s inability to increase glutamate release, the increase in extracellular adenosine induced by the drug seems to preferentially affect only A_1_Rs and not be of sufficient concentration to activate A_2A_Rs. An additional explanation for the preferential activation of A_1_Rs vs. A_2A_Rs with dipyridamole is the evidence for nearer co-localization of A_1_R with equilibrative nucleoside transporters in specific cellular microdomains [35]. Conversely, accumulating evidence indicates that, in other microdomains, the activation of A_2A_Rs by endogenous adenosine requires a particular pool of extracellular adenosine originated from the CD73-mediated formation of ATP-derived extracellular adenosine [36,37,38], in accordance with the physical association of CD73 and A_2A_Rs in the striatum [39]. However, we would favor the first explanation for the effect of dipyridamole in cortico-striatal nerve terminals, since A_2A_R-A_1_R heteromerization implies a tight co-localization of both receptors in the same microdomain of the plasma membrane.

On the other hand, selective A_2A_R antagonists, such as SCH-442416 or MSX-3 (pro-drug of the active compound MSX-2), but not KW-6002 (istradefylline), counteract striatal glutamate release induced by electrical or optogenetic stimulation of cortico-striatal neurons [14,30,40,41]. Importantly, most A_2A_R antagonists are inverse agonists, including SCH-442416 and MSX-2 (results in preparation), while only KW-6002 is a neutral antagonist [42], which therefore cannot counteract a constitutive activity of the A_2A_R. Altogether, these findings with increased endogenous adenosine (with dipyridamole) and exogenous A_2A_R ligands are compatible with the constitutive activity of A_2A_Rs (not forming heteromers with A_1_Rs) playing a more significant role in modulating (enhancing) the basal excitability of cortico-striatal terminals than its activation by endogenous adenosine. On the other hand, endogenous adenosine mostly plays a negative modulation of the excitability of cortico-striatal terminals by preferentially acting on A_1_Rs (forming or not forming heteromers with A_2A_Rs). By promoting a preferential adenosine-mediated presynaptic activation of A_1_Rs vs. A_2A_Rs, dipyridamole could be considered as an indirect agonist of striatal presynaptic A_1_Rs localized in cortico-striatal terminals.

As briefly reviewed in the Introduction, BID in rodents constitutes a model of RLS with construct validity. Rats with BID show an increased sensitivity of cortico-striatal synapses [14], which could be mechanistically related to the well-established cortical motor hyperexcitability of RLS patients [43,44,45]. The present study provides further support for a role of adenosine signaling in this increased sensitivity, more specifically, for a change in A_1_R/A_2A_R stoichiometry in favor of A_2A_Rs in cortico-striatal glutamatergic terminals. In fact, previous experiments had already shown a decrease in A_1_R density in the cortex and striatum and an increase in A_2A_R density in the striatum of rodents with BID [8,9]. These neurochemical changes have not yet been reported in patients with RLS, which would further validate the rodent with BID as an animal model of RLS. The “adenosine hypothesis” of RLS, however, is supported by the efficacy of dipyridamole in treating RLS symptoms [6,7].

Co-localization of A_1_Rs and A_2A_Rs is not specific for the glutamatergic terminals of the striatum and changes in A_1_R/A_2A_R stoichiometry in favor of A_2A_Rs in glutamatergic terminals of other brain areas can also be involved in brain conditions other than RLS. For instance, there is compelling evidence from animal models of a cortical and hippocampal upregulation of A_2A_Rs, without a concomitant upregulation of A_1_Rs, in glutamatergic synapses upon aging and Alzheimer’s disease (AD) [46,47,48,49]. We propose that the relief of A_1_R-mediated inhibition of the constitutive activity of the A_2A_R in the A_1_R-A_2A_R heteromer represents a common mechanism involved in neuropsychiatric conditions with increased glutamatergic transmission. Therefore, blocking the constitutive activity of presynaptic A_2A_Rs with selective inverse agonists or increasing the activation of presynaptic A_1_Rs with a preferential indirect A_1_R agonists like dipyridamole could provide valuable therapeutic approaches for those neuropsychiatric disorders.

## 4. Materials and Methods

### 4.1. Animals

All experiments were performed in accordance with the local animal welfare committee (Centre for Neuroscience and Cell Biology, University of Coimbra, Portugal), European Union guidelines and the Federation of Laboratory Animal Science Associations (FELASA) and were approved by the Animal Care Committee (ORBEA) of the Center for Neuroscience and Cell Biology of the University of Coimbra, Coimbra, Portugal (license number 257). Sprague-Dawley rats were purchased from Charles-River (Écully, France) for breeding. Animals throughout the study were housed with 12 h light on/off cycles under controlled temperature (23 ± 2 °C) and *ad libitum* access to food and water. All efforts were made to minimize the number of animals used and to minimize their stress and discomfort. On post-natal day 16 (PND16), the food of the lactating dams was changed from regular rat chow to a rat chow containing 48 ppm or mg/kg Fe^2+^ (code: TD.80396; ssniff Spezialdiäten GmbH, Soest, Germany). On PND21, the litter was weaned, and 19 male pups were randomly assigned to the adjusted control group (receiving chow with 48 ppm Fe^2+^) and BID group (receiving the modified TD.80396 diet containing residual, 6–8 ppm Fe^3+^). Two to three pups were housed per cage.

### 4.2. Determination of Peripheral Iron Deficiency and Bid

28 days after the initiation of the iron diet, rats were weighed, then deeply anesthetized with halothane in a chamber (no reaction to tail pinch or handling, while still breathing), and decapitated with a stainless-steel guillotine for immediate tissue collection in ice-cold sucrose solution. Some of the dissected brain parts were snap-frozen in liquid nitrogen, while the striata were used instantly for the preparation of purified synaptosomes (see below), which were then snap-frozen and stored at −80 °C until use within a month. In addition, the blood was collected in tubes with 0.5 M K_2_EDTA to perform supplementary hematological analysis by a local external laboratory (Laboratory of Beatriz Godinho, Coimbra, Portugal).

### 4.3. Purified Synaptosomes

#### 4.3.1. Preparation of S1 Fraction

Synaptosome preparation and purification was carried out according to the protocol described by Dunkley et al. [50], with some changes introduced by us [25]. The brains were quickly collected in an ice-cold solution of 320 mM sucrose, 1 mM EDTA and 5 mM Tris, pH 7.4. The pair of striata were isolated and homogenized, using a Potter Teflon, in 6 mL sucrose. This homogenate was divided into three 2 mL Eppendorf tubes, and centrifuged at 4000 *g* for 5 min. The first supernatants (S1) of the first two low-speed centrifugations were layered on top of the discontinuous Percoll gradient for purification.

#### 4.3.2. Discontinuous Percoll Gradient

Percoll gradients (Percoll diluted in the above sucrose solution to 3%, 10%, 15% and 23%) were prepared as detailed before [25,50]. In 15 mL centrifuge tubes, 2 mL of the S1 synaptosomal fraction were layered gently with a peristaltic pump on top of the Percoll gradients. The gradients were centrifuged at 25,000 *g* for 11 min at 4 °C. The purified synaptosomes were removed between the 15% and 23% layers and subsequently, diluted to 15 mL with HEPES buffered medium (HBM) with the following constitution: 140 mM NaCl, 5 mM KCl, 5 mM NaHCO_3_, 1.2 mM NaH_2_PO_4_, 1.2 mM MgCl_2_, 10 mM glucose, and 10 mM HEPES, pH 7.4. The samples were centrifuged at 22,000 *g* for 11 min at 4 °C, the pellet was collected, diluted in 2 mL of HBM and centrifuged at 5000 *g* for 11 min at 4 °C and the final pellet was snap-frozen in liquid nitrogen and stored at −80 °C until use.

#### 4.3.3. Immunolabeling and Flow Synaptometry Analysis

Immunochemical labeling was performed according to a method for staining of intracellular antigens [25,51] for flow cytometry. The pellets obtained after purification in gradients were fixed in 1 mL of 0.25% paraformaldehyde in phosphate buffered saline (PBS; 135 mM NaCl, 1.3 mM KCl, 3.2 mM NaH_2_PO_4_, 0.5 mM KH_2_PO_4_ and 10 mM EDTA) for 1 h at 4 °C. After fixation, they were centrifuged at 5000 *g* for 3 min at 4 °C. The pellet was incubated with PBS with 0.2% Tween 20 for 15 min at 37 °C and then centrifuged at 5000 *g*, and subsequently, washed and pelleted with 0.5 mL of PBS for 3 min at 4 °C. The resulting pellet was resuspended and diluted in ~200 µL PBS. For immunolabeling, 5 µL of these resuspended synaptosomes were incubated with 100 µL of primary and secondary antibodies, all diluted in PBS with 2% normal goat serum (Jackson Immunoresearch, West Grove, PA, USA), for 30 min at 4 °C. As primary antibodies, we used rabbit monoclonal anti-synaptophysin (1:300; Synaptic Systems, Goettingen, Germany), mouse monoclonal anti-VGLUT1 (1:100; Synaptic Systems), mouse monoclonal anti-VGLUT2 (1:10,000; Synaptic Systems), rabbit polyclonal anti-A_1_R (1:300; Invitrogen, Walthan, MA, USA) and guinea pig polyclonal anti-A_2A_R (1:30; Frontiers/Nittobo, Tokyo, Japan). As secondary antibodies, we used goat IgG anti-mouse Cy3 (1:200; Jackson Immunoresearch), goat IgG anti-rabbit FITC (1:200; Jackson Immunoresearch) and goat IgG anti-guinea pig Cy5 (1:200; Abcam, Cambridge, UK). The optimal dilution for the primary and secondary antibodies was determined previously (Ferreira et al., 2015). After each incubation, three washes were carried out for 3 min each, with PBS 0.2% Tween 20, at 5000 *g* at 4 °C. Negative controls without primary antibodies were also carried out, containing only synaptosomes and secondary antibodies.

#### 4.3.4. Detection of Synaptosomes and Data Analysis

Labeled synaptosomal pellets were then resuspended in PBS and were analyzed in a FACSCalibur flow cytometer (four channels; Becton, Dickinson and Company, East Rutherford, NJ, USA). The right dilution for each sample was adjusted to work within a count of 300–400 events per second. Approximately 30,000 events were collected for analysis. From earlier electron microscopy studies (see for instance ref. [28]), we inferred that the size of striatal glutamatergic synapses falls predominantly between 0.5 and 2 µm. Thus, with the help of the Invitrogen Flow Cytometry Sub-micron Particle Size Reference Kit, we calibrated the gating of our equipment for this size range (Figure 2A). Data analysis was performed using BD Cell Quest Pro software. Data were plotted in a dual-parameter dot plot to analyze the percent of co-localization through the upper right quadrant. A threshold was set on forward light scatter to exclude debris. To correct for spectral overlap during multicolor flow cytometry experiments, color compensation was performed. The specific labeling of each sample is calculated by subtracting the percentage of labeling the sample with that of the respective controls and with the percentage of PBS debris (Figure 2B).

### 4.4. Western Blotting

To test the effectiveness of the BID protocol on the brain, cortical samples of both the control and the BID group were homogenized and sonicated in 1% SDS, then proteins were quantified with the colorimetric bicinchoninic acid (BCA) assay. Next, the samples were denaturated with SDS sample buffer (500 mM Tris, 600 mM dithiothreitol, 10.3% SDS, 30% glycerol and 0.012% bromophenol) at 70 °C for 20 min. From each sample, 10 µg of protein was loaded into the gels and subsequently separated by polyacrylamide gel electrophoresis (SDS-PAGE), using a 4% stacking gel (4% bis-acrylamide, tris-HCl (0.5 M, pH 6.8), 10% SDS, 10% ammonium persulfate, 1% tetramethylethylenediamine (TEMED) and a 10% resolving gel 10% bis-acrylamide, tris-HCl (1.5 M, pH 8.8), 10% SDS, 10% ammonium persulfate, 1% TEMED), first at 60 V for 15 min and then at 120 V for 60 min. Then, proteins were transferred to nitrocellulose membranes at 0.75 A for 2 h, at 4 °C with agitation, with CAPS solution (*N*-cyclohexyl-3-aminopropanesulfonic acid) buffered solution with methanol (10 mM CAPS; 10% methanol, pH 11.0). The membranes were blocked with 3% bovine albumin serum (BSA; Merck Biosciences, Darmstadt, Germany) in Tris-buffered saline (10 mM Tris; 150 mM NaCl) containing 0.1% Tween-20 (TBS-T) for 1 h at room temperature. The membranes were then incubated with a monoclonal mouse anti-transferrin receptor antibody (1:2000; Invitrogen), overnight at 4 °C. After incubation, membranes were washed for 3 × 5 min in TBS-T and then incubated with secondary antibody (goat anti-rabbit IgG peroxidase conjugated; Thermo Scientific, Walthan, MA, USA) with 3% BSA for 2 h at room temperature. After washing three times for 5 min, the membranes were processed for protein detection using an enhanced chemiluminescence kit (Pierce™ECL Western Blotting Substrate, Thermo Scientific 32106), and the bands visualized on a on a ChemiDoc Plus imaging system (BioRad, Hercules, CA, USA). After stripping, the membranes were reprobed with an anti-β-actin antibody (1:20,000; Merck Biosciences) for normalization of protein density. Quantification of the optical density of the bands was performed using the Image Lab™ software version 6.0.1 (BioRad).

## Figures and Tables

**Figure 1 molecules-27-01489-f001:**
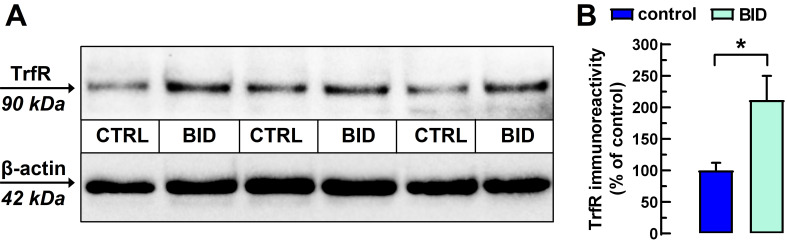
Increased density of transferrin receptor (TrfR) in the cerebral cortex of rats with BID as compared with controls (CTRL). (**A**) Representative blot with 10 µg proteins obtained from total cortical homogenates of three pairs of CTRL and rats with BID. (**B**) TrfR density values were compared to their respective β-actin density values after reprobing the stripped membranes and the average of TrfR/β-actin ratios from CTRL rats were taken as 100%. Statistical comparisons between rats with BID and controls were made with two-tailed unpaired Student’s *t*-test (* = *p* < 0.05).

**Figure 2 molecules-27-01489-f002:**
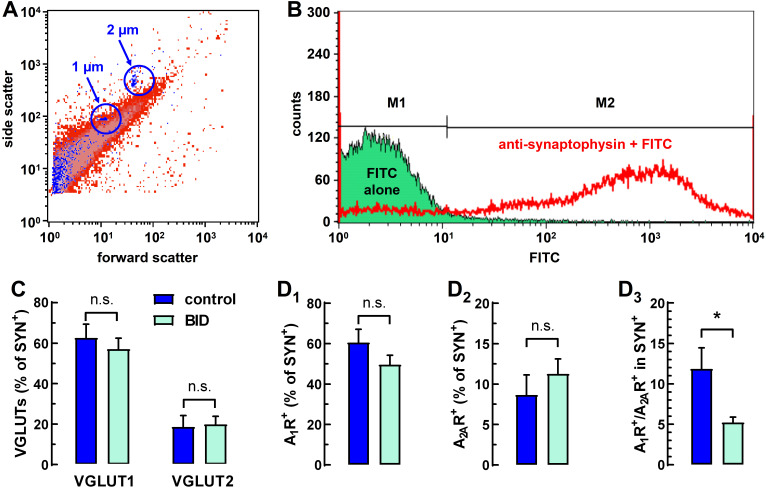
Localization of A_1_R and A_2A_R in striatal terminals (**A**) Representative flow synaptometry dual parameter dot-plot of striatal synaptosomes for size (forward scatter; proportional to the particle size) and for complexity/granularity (side scatter). The logarithmic scales of the x and the y axes represent signal intensity in arbitrary units. Red dots represent FITC-labeled synaptosomes, while blue dots are size calibration beads. (**B**) Representative fluorescence histogram documenting the selectivity of anti-synaptophysin labeling. Specific signal (M2 region) for single-labeled synaptosomes was calculated by subtracting the percentage of labeling by the secondary antibodies alone (histogram filled with green representing synaptosomes incubated only with FITC-conjugated anti-rabbit antibody) from the percentage of labeling by the antibody of interest (histogram in red color). M1 region represents the unlabeled synaptosomes. Note that similar controls were also carried out for the other primary antibodies. (**C**) Striatal presynaptic frequency (as % of synaptophysin positive terminals, SYN^+^) of vesicular glutamate transporters 1 and 2 (VGLUT1/2). (**D_1_**) Percentage of A_1_R^+^ cortico-striatal terminals, (**D_2_**), percentage of A_2A_R^+^ cortico-striatal terminals, and (**D_3_**) ratio between the frequency of inhibitory A_1_R excitatory and the A_2A_R. The inter-animal variability of A_1_R and A_2A_R labelling masks the difference between control and BID in panels D_1_ and D_2_, while in panel D_3_ there is an intra-animal normalization of A_2A_R labelling to A_1_R labelling, better illustrating the effect of BID. For panels D_1_–D_3_, all raw data with the corresponding statistical analyses can be accessed in Appendix A. Bars represent mean + S.E.M. of n = 8–11 animals. Statistical comparisons were made with a two-tailed unpaired Student’s *t*-test (n.s. = not significant; * = *p* < 0.05).

**Figure 3 molecules-27-01489-f003:**
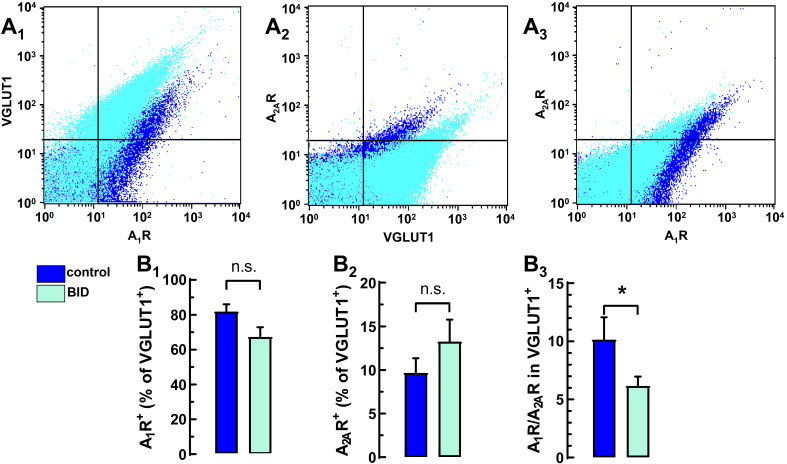
Localization of A_1_Rs and A_2A_Rs in cortico-striatal terminals. (**A_1_–A_3_**) Representative dot-plots showing the colocalization of VGLUT1, A_1_Rs and A_2A_Rs in the upper right quadrants in striatal synaptosomes obtained from a control rat (dark blue) and a BID animal (cyan). Logarithmic scales of the x and the y axes represent the intensity of fluorescence in arbitrary units. Percentage of cortico-striatal terminals (VGLUT1 positive) terminals that are positive for (**B_1_**) A_1_R^+^, (**B_2_**) A_2A_R^+^ and (**B_3_**) ratio between the inhibitory A_1_R and the excitatory A_2A_R. Bars represent mean + S.E.M. of n = 8–11 animals. Statistical comparisons between rats with BID and controls were made with a two-tailed unpaired Student’s *t* test (n.s. = not significant; * = *p* < 0.05).

**Figure 4 molecules-27-01489-f004:**
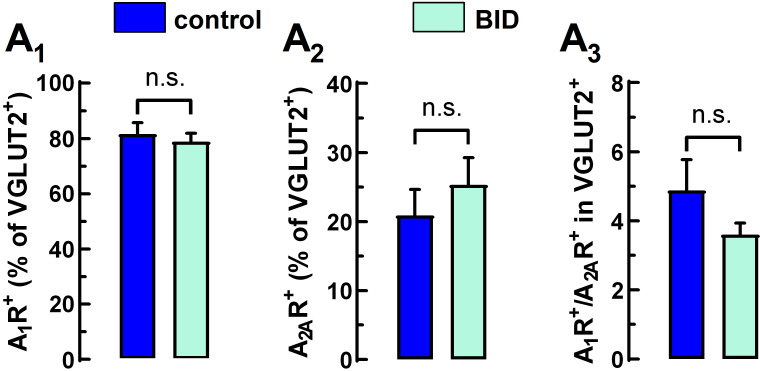
Localization of A_1_Rs and A_2A_Rs in thalamo-striatal terminals. Percentage of thalamo-striatal terminals (VGLUT2 positive) terminals that are positive for (**A_1_**) A_1_R^+^, (**A_2_**) A_2A_R^+^ and (**A_3_**) ratio between the inhibitory A_1_R and the excitatory A_2A_R. Bars represent mean + S.E.M. of n = 8–11 animals. Statistical comparisons between rats with BID and controls were made with a two-tailed unpaired Student’s *t*-test (n.s. = not significant).

**Table 1 molecules-27-01489-t001:** Body weight and haematological parameters.

Parameter	Group	Mean	S.E.M.	*p*
Body Weight (g)	control	296.4	8.47	0.0002
	BID	248.5	6.62	
Erythrocytes (10^12^/L)	control	6.68	0.08	<0.0001
	BID	2.34	0.20	
Hemoglobin (g/dL)	control	14.2	0.23	<0.0001
	BID	5.20	0.31	
Hematocrit (L/L)	control	0.44	0.01	<0.0001
	BID	0.12	0.01	
Mean Corpuscular Volume (fL)	control	65.1	0.54	0.0029
	BID	57.3	2.76	
Mean Globular Haemoglobin (pg)	control	21.1	0.24	0.287
	BID	26.4	2.83	
Mean Corpuscolar Haemoglobin (g/dL)	control	32.4	0.21	0.0093
	BID	44.9	3.16	
Red Cell Distribution Width (%)	control	12.1	0.14	<0.0001
	BID	23.3	2.36	
Leukocytes	control	9.28	1.06	0.0717
	BID	6.43	0.97	
Segmented Neutrophils	control	1.21	0.16	0.8642
	BID	1.09	0.27	
Eosinophils	control	0.08	0.05	0.4814
	BID	0.054	0.03	
Lymphocytes (10^9^/L)	control	7.51	0.81	0.0317
	BID	4.99	0.66	
Monocytes (10^9^/L)	control	0.45	0.11	0.1782
	BID	0.29	0.09	
Thrombocytes (10^9^/L)	control	777.0	26.6	0.0215
	BID	1247.6	131.1	
Serum Iron Concentration (μmol/L)	control	44.1	2.78	<0.0001
	BID	4.39	0.25	

Statistical comparisons between BID rats and controls were made with an unpaired Student’s *t*-test.

## Data Availability

Not applicable.

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
