# Peer review of "Brain Iron Deficiency Changes the Stoichiometry of Adenosine Receptor Subtypes in Cortico-Striatal Terminals: Implications for Restless Legs Syndrome"

_molecules, 2022, doi:10.3390/molecules27051489_

Round 1

Reviewer 1 Report

This manuscript is about interesting topic of molecular mechanisms of brain iron deficiency (BID) as primary pathophysiological mechanism of restless legs syndrome (RLS). BID-rodent model of RLS has been used for study of adenosine receptor subtypes in brain areas of interest and selective decrease in the A1R/A2AR ratio in cortico-striatal terminals has been observed. Study design and methodology are contemporary and adequate. Moreover newly developed FACS-based synaptometric analysis to compare the relative abundance on A1Rs and A2ARs in cortico-striatal and thalamo-striatal glutamatergic terminals has been applied. The results are clearly presented, well documented and critically discussed, and conclusions emerge from them. This study clarifies molecular mechanisms of action of BID in RLS but results are of wider importance, because changes in A1R/A2AR stoichiometry in favor of A2AR in glutamatergic terminals of other brain areas can also be involved in brain conditions other than RLS, such as aging or Alzheimer disease. So results of this study offer valuable new therapeutic options for those disorders / conditions, by blocking the constitutive activity of presynaptic A2AR or increasing the activation of presynaptic A1R. Having in mind all the above I truly recommend this manuscript for the publication in its present form.    

Author Response

We are very pleased by the comments of the reviewer that accepted our manuscript in its original form

Reviewer 2 Report

The manuscript of Rodrigez et al. investigated the effect of brain iron deficiency (BID) on the prevalence of the two striatal postsynaptic adenosine receptor isoforms (A1 and A2). Authors employ a rat iron deficiency nutrition model for in vivo experimentations and analyse A1 and A2 receptor density on isolated striatal neuronal terminals by immunolabelling and flow synaptometry. Authors demonstrate a decrease in the ratio of the inhibitory A1R form in the A1R-A2R complex specifically in the cortico-striatal synapses but not in the thalamo-striatal terminals. Authors argue that this change could provide and explanation for the previously reported enhanced sensitivity of cortico-striatal glutamatergic terminals in BID. Authors suggest that that the observed mechanism (the release of A1R-mediated inhibition of the constitutive activity of the A2AR in the A1R-A2AR heteromer) could represent a common mechanism involved in other neuropsychiatric conditions linked to increased glutamatergic transmission.

The manuscript is written in a very clear fashion, the results are presented in 4 multipanel figures that support the conclusions of the manuscript.

There are some points however, that need to be clarified before the manuscript could be considered for publication.

  1. Page 5, Line 193 (Figure legend for Figure 2D3)“ (D3) ratio between the frequency of excitatory A2AR and the inhibitory A1R”.

 This phrase should be changed for “(D3) ratio between the frequency of inhibitory A1R and the excitatory A2AR” that reflect better the data presented in the graph.

  1. Figure 2D3 seems to show an exaggerated (approximately 3 times) difference between the control and the BID groups based upon the relatively small differences presented in Fig. 2D1 and Fig 2D2. It is even more striking as similar differences presented in Fig. 3 only resulted in approximately two times differences in Fig. 3B3. Authors should verify the data presented in Fig. 2D3. It might be also helpful to present the exact numbers used for the calculations in a supplementary Table.

  1. Page 6, Line 214 (Figure legend for Figure 3B3)“ (D3) ratio between the frequency of excitatory A2AR and the inhibitory A1R” .

 This phrase should be changed for “(D3) ratio between the frequency of inhibitory A1R and the excitatory A2AR” that reflect better the data presented in the graph.

  1. Page 6, Line 215. A panel (D) is referenced which is not included in Figure 3, should be removed.

  1. Page 8, Paragraph: “Animals”. Authors should state the license number of the animal experimental protocol and the name of the appropriate veterinary ethical committee that accepted the protocol.

Author Response

We are grateful to the reviewer for noticing the small inconsistencies in the original manuscript.   Answer to comments on the references to figures: The 3 instances of erroneous reference to the figures "ratio between the frequency of excitatory A2AR and the inhibitory A1R”  as well as the reference to the non-existing "panel D" have been corrected accordingly.   Comments on data and statistical analysis in panels D1-D3 in Figure 2: Figure 2D3 seems to show an exaggerated (approximately 3 times) difference between the control and the BID groups based upon the relatively small differences presented in Fig. 2D1 and Fig 2D2. It is even more striking as similar differences presented in Fig. 3 only resulted in approximately two times differences in Fig. 3B3. Authors should verify the data presented in Fig. 2D3. It might be also helpful to present the exact numbers used for the calculations in a supplementary Table.   Answer: We submit Supplementary Information including the data for Figure 2 panels D1, D2 and D3. Below this table, we also present the statistical analysis obtained by GraphPad Prims 8 for each figure. We also noted that we have inadvertently left out the smallest value pending as "Excluded" for both the control and the BID rats in Panel 3D, which we have added in the revised version. Nevertheless, as it can be seen when comparing with the original Figure 2, the difference is almost invisible. In the previous Figure, the difference was 2.4-times, and now 2.3-times between the control and BID ratio. To answer the valid question of the exaggerated difference, please note that the in-cohort A1R and A2AR data are not normalized for the mean control value, but more importantly, they reflect inter-animal differences. If we normalized these values to control, we should show a 100% bar for the control A1R and the control A2AR, and clearer differences would be already seen in panels D1 and D2 (still not significant though), but that would take away precious information about the percentage of receptor labeling for each terminal.    In contrast, D3 shows includes a normalization, where A2AR frequency is normalized to the intra-animal A1R frequency. This is why the data appear exaggerated in D3. This is now explained in the legend to Figure 2, to make it also clearer to the reader.   It should also also be mentioned that the results were obtained with three different batches of the Nittobo A2AR antibody with very different labeling sensitivity, as can be seen for the variable numbers in the supplementary table.   Answer to Comments on Ethics Statement and Animal Welfare: We have now included the full Ethics Statement in paragraph 4.1.